# Comparative Metabolomic Responses of Three Rhododendron Cultivars to the Azalea Lace Bug (*Stephanitis pyrioides*)

**DOI:** 10.3390/plants13182569

**Published:** 2024-09-13

**Authors:** Bei He, Yuan Zhou, Yu Peng, Dongyun Xu, Jun Tong, Yanfang Dong, Linchuan Fang, Jing Mao

**Affiliations:** 1Institute of Forestry and Fruit Tree, Wuhan Academy of Agricultural Sciences, Wuhan 430070, China; hebei24712@163.com (B.H.); zhouyuan@wuhanagri.com (Y.Z.); 13437100238@163.com (Y.P.); dongdong791123@163.com (D.X.); vivian7484@126.com (J.T.); 18971351076@163.com (Y.D.); 2Horticulture and Forestry College, Huazhong Agricultural University, Wuhan 430070, China

**Keywords:** rhododendron, azalea lace bug, metabolomic, pest resistance, phenylalanine metabolic pathway

## Abstract

Rhododendron, with its high ornamental value and ecological benefits, is severely impacted by the azalea lace bug (*Stephanitis pyrioides*), one of its primary pests. This study utilized three Rhododendron cultivars, ‘Zihe’, ‘Yanzhimi’, and ‘Taile’, to conduct a non-targeted metabolomic analysis of leaf samples before and after azalea lace bug stress using headspace solid-phase microextraction combined with gas chromatography–mass spectrometry (HS-SPME/GCMS) and liquid chromatography–mass spectrometry (LCMS). A total of 81 volatile metabolites across 11 categories and 448 nonvolatile metabolites across 55 categories were detected. Significant differences in metabolic profiles were observed among the different cultivars after pest stress. A total of 47 volatile compounds and 49 nonvolatile metabolites were upregulated in the most susceptible cultivar ‘Zihe’, including terpenes, alcohols, nucleotides, amino acids, and carbohydrates, which are involved in energy production and secondary metabolism. Conversely, ‘Yanzhimi’ showed a downtrend in both the differential volatiles and metabolites related to purine metabolism and zeatin biosynthesis under pest stress. The resistant cultivar ‘Taile’ exhibited moderate changes, with 17 volatile compounds and 17 nonvolatile compounds being upregulated and enriched in the biosynthesis of amino acids, pentose, glucuronate interconversions, carbon metabolism, etc. The phenylalanine metabolic pathway played an important role in the pest resistance of different susceptible cultivars, and relevant metabolites such as phenylethyl alcohol, methyl salicylate, and apigenin may be involved in the plant’s resistance response. The results of this study provide a new perspective on the metabolomics of Rhododendron–insect interactions and offer references for the development of pest control strategies.

## 1. Introduction

Rhododendron (*Rhododendron* L.) plants are important ornamental flowers worldwide, with significant ecological and commercial value. In the cultivation of Rhododendrons, the *Stephanitis pyrioides* Scott, known as the azalea lace bug (ALB), is one of the main pests. It feeds on the foliage of Rhododendrons, affecting their photosynthesis and energy supply [1,2]. Currently, the control of ALBs mainly relies on traditional chemical pesticides, which are found to be inefficient or have potential environmental impacts [3,4].

Previous studies have shown that plants have developed diverse strategies to cope with the attack of herbivorous insects, such as the recognition of feeding signals from pests, internal signal transduction, and the synthesis and release of varied defensive metabolites [5,6]. These metabolites include primary metabolic substances that provide plants with energy and nutrition, such as amino acids, organic acids, and carbohydrates [7]. They may also include secondary metabolites from terpenes and the phenylpropanoid metabolic pathway, including volatile organic compounds (VOCs), like D-limonene, methyl salicylate (MeSA), or phenylethyl alcohol [8,9,10,11,12]. At present, studies on Rhododendrons mainly focus on the substances of flower color and fragrance, as well as the metabolites under certain environmental stresses—see the following examples: anthocyanins, the main substances determining different flower colors in *Rhododendron schlippenbachii*; benzene propanol substances, including benzaldehyde as the main component of the fragrance emitted by *Rhododendron* ‘Nova Zembla’ and *Rhododendron fortunei*; and soluble sugar compounds which can assist in resisting water stress by regulating the osmotic action in *Rhododendron delavayi* [13,14,15]. Few papers are available on the metabolic defense against pest stress in Rhododendrons. The Rhododendron family has abundant germplasm resources; thus, there should be some cultivars with strong resistance to pests [2]. However, the metabolic characteristics of different resistant Rhododendron cultivars under pest stress have not been comprehensively analyzed.

Metabolomics is an omics approach used in the study of plants to explore the resistance mechanisms in biotic stress. The objective of this study was to investigate the metabolic profile of different Rhododendron cultivars with varied susceptibility to ALBs and to provide a comparable analysis of metabolic variation after ALB infestation of leaves. This work may not only represent an interesting insight for future studies in pest resistance of Rhododendron species but also serve as core information for metabolomics comparisons in related research. Exploring the potential key metabolites and metabolic pathways in response to pest stress will offer an important reference for both the further breeding and the pest management of Rhododendron.

## 2. Results

### 2.1. Comparison of ALB Susceptibility among Different Rhododendron Cultivars

Field observations indicated significant differences in the average number of ALBs on the leaves of three cultivars (Figure 1A). ‘Zihe’ had the highest average number of bugs per leaf at 6.58, followed by ‘Yanzhimi’ at 4.08; ‘Taile’ had a significantly lower number than the other two cultivars, with only 0.43 bugs per leaf.

The leaves of ‘Zihe’, ‘Yanzhimi’, and ‘Taile’ were compared pairwise to observe the choices of the ALBs between each pair of cultivars (Figure 1B–D). When the ALBs were placed between the ‘Zihe’ and ‘Taile’ leaves, at 1 h, 66% of the insects were wandering outside the two leaves. Gradually, the insects crawled onto the leaves; by 24 h, an average of 61% of bugs were found to be feeding on the ‘Zihe’ leaves. When the ‘Yanzhimi and ‘Taile’ leaves were provided, 51% of bugs made the choice to move to the ‘Yanzhimi’ leaves within 2 h, and the number of bugs increased to 7.0 at 24 h. Between the ‘Zihe’ and ‘Yanzhimi’ leaves, at 2 h of inoculation, 25% of insects stayed on ‘Yanzhimi’, and 41% chose ‘Zihe’. Subsequently, at 24 h, 64% of bugs fed on ‘Zihe’. From these observations, it can be speculated that, among the three Rhododendron cultivars, ALBs showed the highest preference for ‘Zihe’ and the lowest preference for ‘Taile’.

### 2.2. Identification of Metabolites in Different Rhododendron Cultivars

Qualitative and quantitative analyses of the metabolites by GCMS revealed a total of 81 volatile compounds across 11 categories, namely 29 terpenes, 12 aromatics, 9 alcohols, 6 aldehydes, 6 alkanes, 6 heterocyclic compounds, 6 esters, 4 ketones, 1 olefin, 1 ether, and 1 amine (Figure 2A).

In total, 448 compounds belonging to 55 classes were identified in LCMS, including 63 flavonoids, 47 amino acids, 45 organic acids, and 34 types of carbohydrates (Figure 2B). A further 170 compounds were found to be distributed across 45 subclasses, such as polyphenols, plant hormones, nucleic acid derivatives, vitamins, and fatty acids.

### 2.3. Differential Accumulated Metabolites after ALB Infestation

#### 2.3.1. Volatile Differential Accumulated Metabolites in GCMS

The overall distribution of the differential accumulated metabolites (DAMs) of volatiles after ALB infestation is presented in Figure 3. In the comparison group of ‘Zihe’, the number of upregulated metabolites was the highest: 47 differential volatile compounds were identified, comprising 21 terpenes, 7 aromatics, 6 alcohols, 3 esters, and 10 other substances distributed among different classes (Figure 3A,B). In ‘Yanzhimi’-infested leaves, only three differential volatiles were detected, including two alkanes and one terpene, all of which were downregulated. ‘Taile’ showed the upregulation of 17 differential volatiles and downregulation of 2 compounds; additionally, it showed a significant abundance of changes in terpenes, aromatics, and alcohols.

Further analysis using a Venn diagram revealed (Figure 3C) that 33 DAMs specialized in ‘Zihe’, such as maltol, D-limonene, and Neoabietadiene, were significantly upregulated (Table 1). Moreover, ‘Zihe’ had 14 DAMs in common with ‘Taile’; here, 13 compounds were upregulated, including phenylethyl alcohol, benzyl alcohol, and methyl salicylate (Figure 3D). 

#### 2.3.2. Nonvolatile DAMs in LCMS

Metabolic profiling under ALB stress revealed significant differences among the three Rhododendron cultivars (Figure 4A). ‘Zihe’ produced 52 DAMs, with only 3 being downregulated. The other 49 DAMs were all upregulated, mainly classified into carbohydrates, organic acids, nucleotides, and amino acids. Flavonoids, polyphenols, phenylpropanoids, and other secondary metabolites were also found to be significantly upregulated in ‘Zihe’ after pest infestation (Figure 4C).

‘Taile’ exhibited 21 DAMs, with 4 compounds being significantly downregulated. The other 17 DAMs were upregulated; these could mostly be categorized as organic acids, amino acids, and carbohydrates (Figure 4D). 

‘Yanzhimi’ had 25 DAMs with 14 upregulated compounds and 11 downregulated compounds. The upregulated compounds were predominantly flavonoids, such as isovitexin and engeletin. Nucleotides, amino acids, and organic acids were primarily downregulated (Figure 4E).

A Venn diagram (Figure 4B) showed that ‘Zihe’ and ‘Taile’ had three compounds in common: dulcitol, creatine, and d-xylulose. These were all upregulated after infestation. ‘Taile’ and ‘Yanzhimi’ had two metabolites in common: adenosine (upregulated in ‘Taile’ and downregulated in ‘Yanzhimi’) and δ-Hexanolactone (downregulated in both). ‘Zihe’ and ‘Yanzhimi’ had the same three upregulated DAMs: engeletin, kaempferol 7-O-β-D-glucopyranoside, and N7-methylguanosine.

### 2.4. The KEGG (Kyoto Encyclopedia of Genes and Genomes) Annotation Analysis of DAMs

#### 2.4.1. The KEGG Annotation of Volatile DAMs

The results of pathway annotation for GCMS data suggested that, after ALB stress, ‘Zihe’ was enriched in 11 types of pathways, including polyketide sugar unit biosynthesis, monoterpenoid biosynthesis, and phenylalanine metabolism. Relevant substances such as 1,3-Propanediamine, (+)-delta-Cadinen, L-.alpha.-Terpineol, and phenylethyl alcohol were also upregulated. ‘Taile’ had differential metabolites that were enriched in five metabolic pathways, namely phenylalanine metabolism, sesquiterpenoid and triterpenoid biosynthesis, metabolic pathways, and the biosynthesis of secondary metabolites and monoterpenoids. No significant enrichment of the pathways was noted for ‘Yanzhimi’ when carrying out a comparison of before and after the inoculation of ALBs (Table 1).

#### 2.4.2. The KEGG Annotation of Nonvolatile DAMs

Under the stress of ALBs, DAMs were upregulated in ‘Zihe’, enriched in 11 pathways, including the following: carbon metabolism; the interconversions of pentose and glucuronate; phenylalanine, tyrosine, and tryptophan biosynthesis; isoflavonoid biosynthesis; and alanine, aspartate and glutamate metabolism. The compounds involved in these processes include D-Glutamic acid, L-Phenylalanine, Apigenin, and γ-Aminobutyric acid (Table 2).

‘Taile’ was found to demonstrate the enrichment of DAMs in 10 metabolic pathways, including the biosynthesis of amino acids, pyruvate metabolism, carbon fixation in photosynthetic organisms, and phosphonate and phosphinate metabolism. The metabolites involved include D-Glutamic acid and D-Xylulose—categorized as carbohydrates and amino acids—which were significantly upregulated.

‘Yanzhimi’ showed enrichment in purine metabolism, zeatin biosynthesis, and the metabolic pathway. Relevant compounds, including adenosine, adenine, sinapic acid, and D-Proline, were all found to be downregulated under the stress of ALBs.

## 3. Discussion

### 3.1. Metabolic Profiles of Different Rhododendron Cultivars in Response to Insect Herbivory

After ALB infestation, distinct volatile metabolic responses were observed among the three Rhododendron cultivars. The most susceptible cultivar, ‘Zihe’, exhibited a heightened production of terpenes, aromatics, and alcohol compounds compared to the other two cultivars. Specifically, ‘Zihe’ exhibited elevated accumulation of maltol, D-limonene, and Neoabietadiene. Maltol has been suggested to play a role in the biosynthetic pathways of insect hormones, potentially impacting their development and reproduction. Specifically, it has been observed that elevated concentrations of maltol can lead to mortality in spruce budworm (*Choristoneura fumiferana* Clemens) [16]. D-limonene possessed potent insecticidal properties against the cowpea weevil (*Callosobruchus maculatus* Fabricius) and exhibited cytotoxic and repellent effects against a range of deleterious insects [17,18]. Neoabietadiene, belonging to the abietane class, participated in the flow and secretion of the resin, thereby protecting the trees from the invasion of pests and diseases [19]. In addition, ‘Zihe’ regulated 14 differential volatile compounds after an infestation of ALBs; these were in common with those regulated by the resistant cultivar ‘Taile’. These results indicated that the promotion of these volatiles is important in inducing the ability to combat ALB attacks in Rhododendron.

Moreover, in this study, ‘Zihe’ predominantly increased the biosynthesis of primary metabolites, and ‘Taile’ demonstrated a moderate change, whereas ‘Yanzhimi’ downregulated both the purine metabolism and zeatin biosynthesis under pest stress. Primary metabolites, including amino acids, carbohydrates, organic acids, and nucleotides, supplied nutrition and energy for the plant to maintain normal growth; additionally, they served essential functions as precursors for the synthesis of metabolites and signaling molecules that are associated with stress caused by insect herbivores [7,20,21,22]. Purine metabolism involves the synthesis and degradation of purine nucleotides, which are essential for the production of energy, signaling molecules, and genetic material. During stress, the modulation of purine levels can influence the production of secondary metabolites, which are vital for a plant’s immune response and adaptation to stress [23]. On the other hand, zeatin biosynthesis is a key component of cytokinin hormone production, which is vital for regulating plant growth and development. Moreover, it has been reported to modulate the expression of defense-related genes, thereby influencing a plant’s tolerance to stress [24]. Therefore, it can be speculated—as reported for other plants under biotic stress—that the resistant cultivar ‘Taile’ possesses the ability to stabilize primary metabolism to maintain normal physiology and produce pest-repellent substances. In contrast, the diminished capacity for energy substance synthesis in ‘Yanzhimi’ may account for its susceptibility to insects [25,26,27]. However, ‘Zihe’ combated pest infestation by stimulating primary metabolisms, which probably produced more sugar or other nutrients that are attractive to ALBs, indicating that resistance can be achieved through complex interactions within herbivores through reprogramming the host metabolic profiles to acquire more nutrients or evade the host’s defense [25,28]. 

Furthermore, ‘Zihe’ exhibited a significant accumulation of D-glutamic acid, which functioned as a nitrogen assimilation regulator and can be transformed into other amino acids, such as proline, through a series of biochemical reactions [21]. Glutamic acid could also operate as a signal to modulate plant growth, development, and defense mechanisms; additionally, it was found to be a precursor to γ-aminobutyric acid (GABA), which could further participate in the biosynthesis of salicylic acid [29]. Notably, the accumulation of GABA in ‘Zihe’ was also detected. It has been proven that GABA, as a neurotransmitter, could disrupt insect feeding by inhibiting insect neuronal γ-aminobutyric acid-targeted Chloride channels [30]. During stress, GABA could regulate plant development and activate defense responses as a signaling molecule [31]. This finding suggested that, although ‘Zihe’ was the most attractive to ALBs, it could develop a higher tolerance against feeding stress through the modulation of primary metabolites. 

### 3.2. The Pivotal Role of the Phenylalanine Metabolism in Rhododendron Insect Resistance

The phenylalanine pathway is intricately linked with various defense mechanisms in plants, providing a multi-layered approach to protection against biotic stresses. Both ‘Zihe’ and ‘Taile’ exhibited an enrichment of the phenylalanine metabolic pathway (Figure 5). The downstream substance phenylethyl alcohol was significantly increased in ‘Zihe’ and ‘Taile’. Phenylethyl alcohol was also detected in ‘Yanzhimi’ (Appendix A). Studies have reported that phenylethyl alcohol can inhibit the mycelial growth and conidia germination of *Botrytis cinerea*. Furthermore, natural phenylethyl alcohol has demonstrated efficacy in the control of potato late blight by blocking the oxidative phosphorylation pathway in *Phytophthora infestans* [32,33]. Phenylethyl alcohol potentially plays a role in plant defense mechanisms by acting as a signal molecule that triggers the activation of defense genes and the development of systemic acquired resistance, which offers sustained protection against pathogens; alternatively, due to its antimicrobial properties, phenylethyl alcohol can strengthen structural barriers to inhibit the growth of pathogens [34]. Based on these findings, it is speculated that phenylethyl alcohol may help in inducing the defense mechanism against fungal infections in Rhododendrons that are potentially induced by ALBs.

The phenylalanine pathway intersects with the hormonal signaling pathways involved in plant defense. For instance, salicylic acid (SA), a plant hormone, can induce phenylalanine ammonia-lyase (PAL) gene expression and is involved in systemic acquired resistance [35]. Methyl salicylate (MeSA), also derived from phenylalanine, is an important metabolic substance that enhances the stress resistance of plants. Many studies have demonstrated that, in response to pest infestation, the primary volatile organic compound emitted by plants was methyl salicylate. This compound can improve plant resistance by repelling and inhibiting the activity of insects, as well as attracting natural enemies [10,36,37]. An increase in MeSA content was noted across all three Rhododendrons, with a particularly significant elevation in ‘Zihe’ and ‘Taile’, potentially contributing to the observed variability in insect resistance among the cultivars. 

In addition, naringenin was detected in ‘Zihe’, and its downstream substance, apigenin, showed significant upregulation. Similarly, a modest increase in apigenin was noted in ‘Yanzhimi’ under insect stress. Both naringenin and apigenin are important flavonoid compounds with phenylalanine as their initial substrate. These flavonoids are recognized for their significant contributions to plant defense mechanisms against biotic stress. Specifically, naringenin and its derivatives have been shown to influence the feeding behavior of *Myzus persicae* and to impair the cognitive actions of honeybees (*Apis mellifera*). Moreover, the application of exogenous apigenin has been demonstrated to enhance wheat’s (*Triticum aestivum* L.) resistance to Fusarium head blight (*Fusarium graminearum*) [38,39,40]. Apigenin may act as a chemical barrier that deters insect feeding or disrupts their digestion, thereby reducing the damage caused by herbivores; its strong antioxidant capacity could potentially help plants mitigate oxidative stress from insect infestation [41]. This suggests that these flavonoid compounds may play a crucial role in the defense response of Rhododendrons to pest stress.

To summarize the above, the phenylalanine metabolic pathway serves as a critical defense component in plants. Within Rhododendrons, the phenylalanine metabolic pathway is instrumental in the generation of compounds such as phenylethyl alcohol, methyl salicylate, and apigenin, which are pivotal in the plant’s defense against the stress caused by ALBs. These bioactive metabolites are integral to the plant’s resistance and provide a multifaceted response to biotic challenges.

## 4. Materials and Methods

### 4.1. Materials

The experimental Rhododendron plants were sourced from the germplasm resource nursery at the Wuhan Academy of Agricultural Sciences. Three cultivars were selected for this study: ‘Yanzhimi’ (YZM), ‘Zihe’ (ZH), and ‘Taile’ (TL). The test Rhododendron plants had canopies measuring approximately 50 cm by 60 cm and were grown in 3-gallon plastic pots; the potting mix consisted of peat, bark, and perlite (1:1:1); all potted seedlings were uniformly maintained on a nursery bed.

Experimental azalea lace bugs (ALBs) were collected from the Rhododendron germplasm resource nursery at the Wuhan Academy of Agricultural Sciences and then exposed to a 14/10 h (light/dark) photoperiod at 25 ± 1 °C in an artificial climate chamber; the relative humidity was controlled at 80%.

### 4.2. Methods

#### 4.2.1. Field Investigation of Insect Resistance among the Three Cultivars

During the high-incidence season of the ALBs in May 2023, the number of adult ALBs was noted on mature leaves below the top buds on the branches of four directions (east, south, west, and north), and at least 3 leaves were observed on each branch, with a total of five plants counted for every cultivar. The average number of adult bugs number per leaf was noted for comparison among different cultivars.

#### 4.2.2. Comparison of Feeding Preference among ALBs to the Three Cultivars

Leaf disks with a 15 mm diameter from the middle position along the main vein of the leaves were placed on a glass Petri dish lined with moist filter paper. After starving for 8 h, ten adult ALBs were placed at the center of the Petri dish, where they were allowed to choose and feed between two different cultivars. The number of bugs feeding on the leaves was recorded at 1, 2, 4, 6, and 24 h, with ten replicates for each comparison.

#### 4.2.3. ALB Inoculation and Sample Collection

The Rhododendron subjected to infestation by ALBs is referred to as the SH (abbreviated from ‘*Stephanitis pyrioides* Hurt’) treatment, while the untreated control is designated as CK. Twenty adult bugs, after an 8 h starving period, were randomly selected and inoculated onto the underside of the mature leaves beneath the top buds of Rhododendron and then covered with transparent mesh bags to prevent escape. The mesh was also placed on the control plants. After 24 h, samples were collected, with at least three replicates for each treatment. All collected samples were quickly frozen in liquid nitrogen and stored at −80 °C.

#### 4.2.4. Headspace Solid-Phase Microextraction (HS-SPEM) and GCMS Analysis

Samples were ground to a powder in liquid nitrogen. One gram of the powder was transferred immediately to a 20 mL headspace vial (Agilent, Palo Alto, CA, USA) containing a saturated NaCl solution. The vials were sealed using crimp-top caps with TFE (Thermally Fused Elastomer)–silicone headspace septa (Agilent, Palo Alto, CA, USA). At the time of SPME analysis, each vial was maintained at 60 °C for 10 min, then, a 65 µm divinylbenzene/carboxen/polydimethylsilioxan fiber (Supelco, Bellefonte, PA, USA) was exposed to the headspace of the sample for 20 min at 60 °C. After sampling, desorption of the VOCs (Volatile Organic Compounds) from the fiber coating was carried out in the injection port of the GC apparatus (Model 7890B; Agilent) at 250 °C for 5 min in the split-less mode. The GC-MS conditions are provided in Appendix A. The identification and quantification of compounds were carried out using an Agilent Model 7890B GC-7000D mass spectrometer with the self-built data system library (MWGC) and a linear retention index.

#### 4.2.5. Sample Extraction and HPLC-MS/MS Analysis

Tissues (100 mg) were individually ground with liquid nitrogen, and the homogenate was resuspended with 500 μL prechilled 80% methanol and 0.1% formic acid through a well vortex process. The samples were incubated on ice for 5 min and then centrifuged at 15,000 rpm, 4 °C, for 10 min. The supernatant was diluted to a final concentration containing 53% methanol by LC-MS-grade water; then, it was injected into the LC-MS/MS system using an ExionLC™ AD system (SCIEX, Framingham, MA, USA) coupled with a QTRAP^®^ 6500+ mass spectrometer (SCIEX, Framingham, MA, USA). The details of the liquid phase and MS conditions are provided in Appendix A. The detection of the experimental samples using MRM (multiple reaction monitoring) was based on the Novogene in-house database. 

### 4.3. Statistical Analysis

Before Tukey’s test of significant difference in the average number of ALBs per leaf on 3 cultivars of Rhododendron, the normality of the raw data was analyzed by the Kolmogorov–Smirnov test, followed by a test for homogeneity of variance using Levene’s test and then proceeded with one-way analysis of variance (ANOVA). All these analyses were processed using SPSS Statistics 27.0.1 (IBM, Armonk, NY, USA).

In data analysis of metabolites, unsupervised PCA (principal component analysis) was performed by statistics function prcomp within R (www.r-project.org) (Appendix A). VIP (variable importance in projection—VIP) values were extracted from partial least squares discriminant analysis (PLS-DA) results; these were generated using the R package MetaboAnalystR and MetaX (Appendix A). Differential accumulated metabolites were determined by VIP >= 1 and absolute Log2FC (fold change) >= 1. The identified metabolites were annotated using the KEGG database, the HMDB (Human Metabolome Database) database, and the Lipidmaps database (www.kegg.jp/kegg/compound/, www.genome.jp/kegg/, www.hmdb.ca/, www.lipidmaps.org/, accessed on 27 June 2024). Annotated metabolites were then mapped to the KEGG pathway database (www.kegg.jp/kegg/pathway.html, accessed on 29 June 2024). Pathways with significantly regulated metabolites were then fed into MSEA (metabolite sets enrichment analysis), and their significance was determined by the hypergeometric test’s *p*-values (Appendix A). Excel 2021 software was used to process the average data of the replicates and to construct graphs. Graph Pad Prism 8.4.3 was also used for graph generation. 

## 5. Conclusions

Upon comprehensive analysis, it is evident that the metabolic responses initiated by three distinct insect-susceptible cultivars of Rhododendron under ALB stress exhibit significant differences. Specifically, ‘Zihe’ exhibited a more robust metabolic change in both the primary pathway and the secondary metabolism process, exhibiting a complex interaction with the herbivores. ‘Taile’ could tolerate a moderate fluctuation in metabolic activity, thereby demonstrating resistance to the stress imposed by ALBs. Conversely, ‘Yanzhimi’ possessed a weaker resistance to this insect, potentially due to limitations in the synthesis of energy or nutritional substances in its metabolic processes. The phenylalanine metabolic pathway was determined to have a significant regulatory function in defense against ALBs. Substances, including phenylethyl alcohol, methyl salicylate, and apigenin, were found to be upregulated in all three Rhododendron cultivars. The results could be applied to clarify the metabolic mechanism of Rhododendron plants during ALB infestation and to provide new resources to control this insect.

## Figures and Tables

**Figure 1 plants-13-02569-f001:**
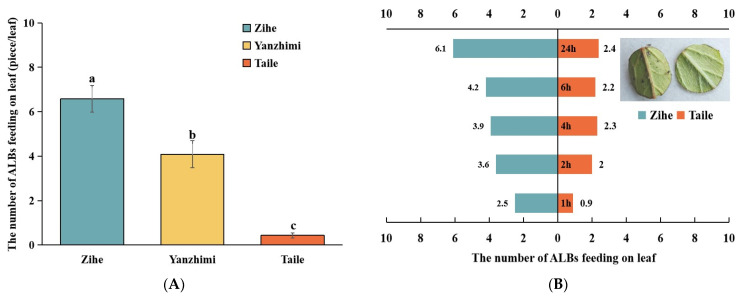
The susceptibility of different Rhododendron cultivars to ALB (azalea lace bug, *Stephanitis pyrioides*). (**A**) The average number of ALBs on the leaves of different Rhododendron cultivars in the field; (**B**) the feeding selection of the ALBs between ‘Zihe’ and ‘Taile’; (**C**) the feeding selection of the ALBs between ‘Yanzhimi’ and ‘Taile’; (**D**) the feeding selection of the ALBs between ‘Zihe’ and ‘Yanzhimi’. Note: different letters indicate statistically significant differences according to Tukey’s test, *p* < 0.05.

**Figure 2 plants-13-02569-f002:**
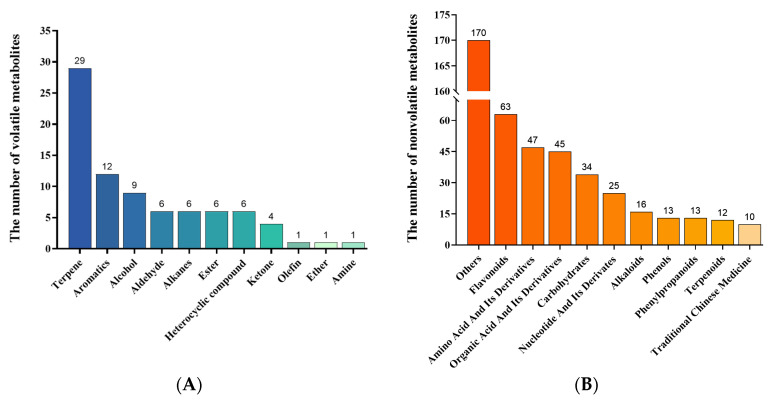
Numbers of all volatile (**A**) and nonvolatile (**B**) metabolites identified in Rhododendron leaf samples.

**Figure 3 plants-13-02569-f003:**
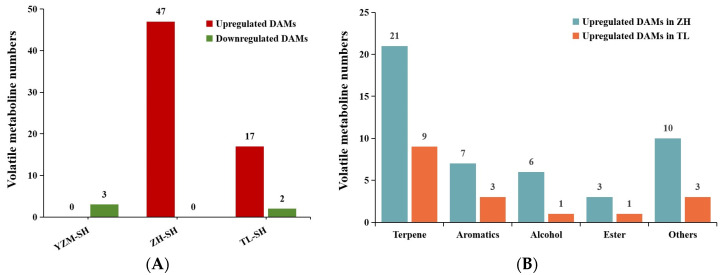
Volatile DAMs (differentially accumulated metabolites) in three Rhododendron cultivars. (**A**) The number of upregulated and downregulated DAMs in 3 cultivars infected by the ALBs (azalea lace bugs); (**B**) the types of DAMs upregulated in‘Zihe’(ZH) and ‘Taile’ (TL); (**C**) Venn diagram of volatile DAMs in 3 cultivars after pest infestation; (**D**) regulation pattern of the overlapped DAMs between ZH and TL. Note: YZM-SH—‘Yanzhimi’ after ALB stress; YZM-CK—‘Yanzhimi’ with no ALB infestation; ZH-SH—‘Zihe’ after ALB stress; ZH-CK—‘Zihe’ with no ALB infestation; TL-SH—‘Taile’ after ALB stress; TL-CK—‘Taile’ with no ALB infestation (below are the same).

**Figure 4 plants-13-02569-f004:**
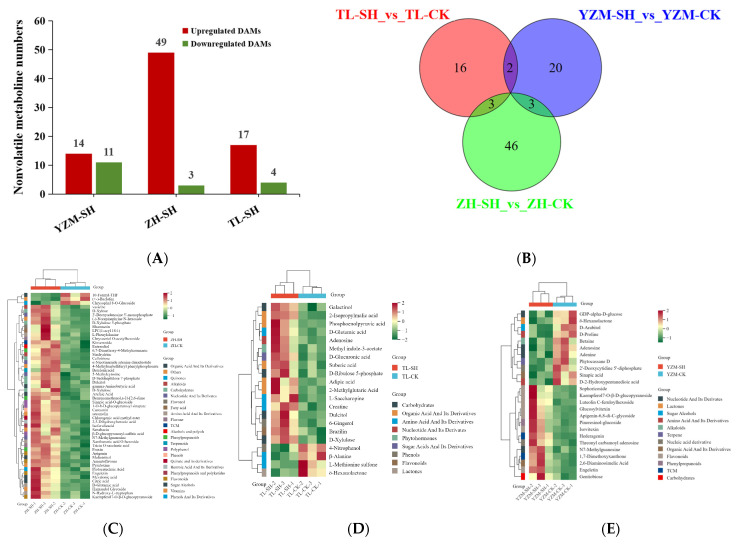
Nonvolatile DAMs (differentially accumulated metabolites) in three Rhododendron cultivars. (**A**) The number of upregulated and downregulated DAMs in three Rhododendron cultivars infected by ALBs; (**B**) Venn diagram of DAMs in three cultivars after pest infestation; (**C**) cluster heatmap of nonvolatile DAMs in ZH(‘Zihe’); (**D**) cluster heatmap of nonvolatile DAMs in TL (‘Taile’); (**E**) cluster heatmap of nonvolatile DAMs in YZM (‘Yanzhimi’).

**Figure 5 plants-13-02569-f005:**
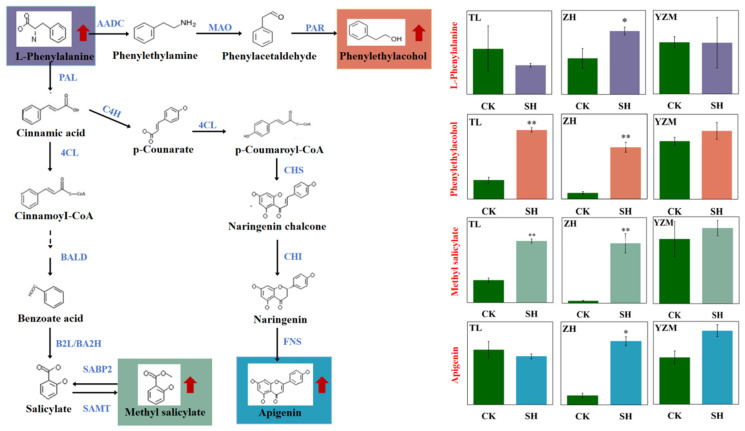
Differential metabolites involved in the phenylalanine metabolism pathway. Note: AADC—aromatic amino acid decarboxylase; MAO—monoamine oxidase; PAR—2-phenylacet aldehyde reductase; PAL—phenylalanine ammonialyase; C4H—cinnamate 4-hydroxylase; 4CL—4-coumarate:CoA ligase; BALD—benzaldehyde dehydrogenase; B2L/BA2H—benzoic acid 2-hydroxylase; SABP2—salicylic acid binding protein 2; SAMT—salicylic acid carboxyl methyltransferase; CHS—chalcone synthase; CHI—chalcone isomerase; FNS—flavone synthase. Asterisks indicate significantly (*, *p* < 0.05) and extremely significantly (**, *p* < 0.01) upregulated differential metabolites.

**Table 1 plants-13-02569-t001:** Enrichment pathways of volatile DAMs (differentially accumulated metabolites) in different Rhododendron cultivars.

Group	Metabolic Pathway	Annotated Differential Metabolites
TL-SH vs. TL-CK	Phenylalanine metabolism	Phenylethyl alcohol ↑
	Sesquiterpenoid and triterpenoid biosynthesis	(+)-delta-Cadinene ↑
	Metabolic pathways	Benzyl alcohol ↑; L-.alpha.-Terpineol ↑; (+)-delta-Cadinen ↑
	Biosynthesis of secondary metabolites	L-.alpha.-Terpineol ↑; (+)-delta-Cadinen ↑
	Monoterpenoid	L-.alpha.-Terpineol ↑
ZH-SH vs. ZH-CK	Polyketide sugar unit biosynthesis	Maltol ↑
	Glycine, serine and threonine metabolism	1,3-Propanediamine ↑
	beta-Alanine metabolism	1,3-Propanediamine ↑
	Monoterpenoid biosynthesis	D-Limonene ↑; L-.alpha.-Terpineol ↑; (1S,5S)-6,6-Dimethyl-2-methylenebicyclo[3.1.1]heptan ↑
	Biosynthesis of secondary metabolites	Maltol ↑; (+)-delta-Cadinen ↑; D-Limonene ↑, etc.
	Limonene and pinene degradation	D-Limonene ↑
	Arginine and proline metabolism	1,3-Propanediamine ↑
	Sesquiterpenoid and triterpenoid biosynthesis	(+)-delta-Cadinen ↑
	Metabolic pathways	D-Limonene ↑; Benzyl alcohol ↑; 1,3-Propanediamine ↑, etc.
	Phenylalanine metabolism	Phenylethyl alcohol ↑
	Diterpenoid biosynthesis	Neoabietadiene ↑
YZM-SH vs. YZM-CK	Nothing	Nothing

Note: ↑ indicates the upregulation of differential annotated metabolite expression; ↓ indicates the downregulation of differential annotated metabolite expression. YZM-SH—‘Yanzhimi’ after ALB stress; YZM-CK—‘Yanzhimi’ with no ALB infestation; ZH-SH—‘Zihe’ after ALB stress; ZH-CK—‘Zihe’ with no ALB infestation; TL-SH—‘Taile’ after ALB stress; TL-CK—‘Taile’ with no ALB infestation.

**Table 2 plants-13-02569-t002:** Enrichment pathways of nonvolatile DAMs in different Rhododendron cultivars.

Group	Metabolic Pathway	Annotated Differential Metabolites
YZM-SH vs. YZM-CK	Purine metabolism	Adenosine ↓; Adenine ↓
	Zeatin biosynthesis	Adenine ↓
	Metabolic pathways	GDP-alpha-D-glucose ↓; Sinapic acid ↓; Betaine ↓; Adenosine ↓; Adenine ↓; D-Arabitol ↓; 2′-Deoxycytidine 5′-diphosphate ↓; D-Proline ↓
TL-SH vs. TL-CK	Biosynthesis of amino acids	Phosphoenolpyruvic acid ↑; D-Ribulose 5-phosphate ↑; 2-Isopropylmalic acid ↑; L-Saccharopine ↑
	Pyruvate metabolism	Phosphoenolpyruvic acid ↑; 2-Isopropylmalic acid ↑
	Pentose and glucuronate interconversions	D-Ribulose 5-phosphate ↑; D-Xylulose ↑
	Carbon fixation in photosynthetic organisms	Phosphoenolpyruvic acid ↑; D-Ribulose 5-phosphate ↑
	Phosphonate and phosphinate metabolism	Phosphoenolpyruvic acid ↑
	Riboflavin metabolism	D-Ribulose 5-phosphate ↑
	Galactose metabolism	Galactinol ↑; Dulcitol ↑
	Lysine biosynthesis	L-Saccharopine ↑
	Carbon metabolism	Phosphoenolpyruvic acid ↑; D-Ribulose 5-phosphate ↑
	Metabolic pathways	Phosphoenolpyruvic acid ↑; D-Ribulose 5-phosphate ↑; Creatine ↑; 2-Isopropylmalic acid ↑; Adenosine ↑; L-Saccharopine ↑; Dulcitol ↑; D-Xylulose ↑; D-Glutamic acid ↑, etc.
ZH-SH vs. ZH-CK	Carbon metabolism	D-Xylulose 5-phosphate ↑; 10-Formyl-THF ↓; Citric acid ↑; D-Sedoheptulose 7-phosphate ↑
	Pentose and glucuronate interconversions	D-Xylulose 5-phosphate ↑; D-Xylose ↑; D-Xylulose ↑
	Metabolic pathways	2,5-Dihydroxybenzoic acid ↑; Apigenin-6,8-di-C-glycoside ↑; Mevalonic acid ↑; Pyridoxine ↑; Dulcitol ↑; D-Glutamic acid ↑; 2′-Deoxyadenosine 5′-monophosphate ↑, etc.
	One carbon pool by folate	10-Formyl-THF ↓
	Terpenoid backbone biosynthesis	Mevalonic acid ↑
	Pentose phosphate pathway	D-Xylulose 5-phosphate ↑; D-Sedoheptulose 7-phosphate ↑
	Phenylalanine, tyrosine, and tryptophan biosynthesis	Protocatechuic Acid ↑; L-Phenylalanine ↑
	Isoflavonoid biosynthesis	Apigenin ↑
	Biosynthesis of amino acids	D-Xylulose 5-phosphate ↑; L-Phenylalanine ↑; Citric acid ↑; D-Sedoheptulose 7-phosphate ↑
	Alanine, aspartate and glutamate metabolism	γ-Aminobutyric acid ↑; Citric acid ↑
	Carbon fixation in photosynthetic organisms	D-Xylulose 5-phosphate ↑; D-Sedoheptulose 7-phosphate ↑

Note: ↑ indicates the upregulation of differential annotated metabolite expression; ↓ indicates the downregulation of differential annotated metabolite expression.

## Data Availability

The data supporting reported results are included in this manuscript and its Appendix A.

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
