# Peer review of "Comparative Metabolomic Responses of Three Rhododendron Cultivars to the Azalea Lace Bug (Stephanitis pyrioides)"

_plants, 2024, doi:10.3390/plants13182569_

Round 1

Reviewer 1 Report

Comments and Suggestions for Authors

Dear authors, please find the PDF attached. 

Comments on the Quality of English Language

Expression can be improved. I suggest using a proofreading aid or consulting a native speaker.

Author Response

Dear reviewer,

     Thank you for your insightful and constructive comments on our manuscript. We have provided a detailed point-by-point response to the your comments. Please see the attachment.

Reviewer 2 Report

Comments and Suggestions for Authors

The authors compared the metabolomic profiles of plants with different susceptibilities to the Azalea Lace Bag. It is important study on the response of plants to insect stress; however, the description of methods and controls could be more precise. First of all, English text is difficult to read. Then, what they did in 4.2.2 needs to be clarified. They introduced larvae on the leaves, and then they counted bugs, which is confusing.

Then, in 4.2.3, do they introduce larvae or adult bugs? Was the mesh also placed on the control leaves? It would be good to describe it with precision in the material methods and in Figure 1. It is important as stress from larvae and adult bug might differ by plants. There is good and interesting data and results, but they need to be clearly described. 

 In Figure 2, TCM needs to be clarified.

In the description of Figure 2A, alkane is missing in the sentence (page 3, first sentence in 2.2)

On page 4, it would be beneficial to clarify the meaning of 'up-accumulated' and 'down-accumulated '. This will ensure that the reader understands these terms and their implications in the context of the research.

On page 9, ...by inhibiting insect neuronal not nerve

4.22 starving

Author Response

(The authors gave the same response as above.)

Reviewer 3 Report

Comments and Suggestions for Authors

Abstract section

Major comments:

1.     The results in the abstract section highlight the detection of 81 volatile and 448 non-volatile metabolites with significant differences in metabolic profiles post-pest stress. While this is impressive, the categorization and functional implications of these metabolites could be better articulated. For instance, reporting the specific pathways (like the phenylalanine metabolic pathway mentioned) is crucial, but the mechanistic detail on how these pathways contribute to the plant’s defense mechanism would offer deeper insights for other reader.

2.     It is commendable that the abstract points out the cultivar-specific responses. Specifically, ‘Zihe’s up-regulation of various metabolites in ‘related to energy production and secondary metabolism, and ‘Taile’s moderate metabolic alterations. However, it would be beneficial to briefly explain it further. Does this up-regulation fail to prevent damage or is it a consequence of significant damage eliciting a greater response? Moreover, the abstract notes ‘Yanzhimi’s downtrend in purine metabolism and zeatin biosynthesis-providing the implications of these trends could elucidate whether these pathways are crucial for resistance or susceptibility.

Minor comments:

1.     Please ensure consistent usage of more technical terms like “up-regulated” and “down-regulated” when discussing metabolic changes, avoiding confusion.

2.     The first sentence is quite long and complex with grammar suspiciousness. For example, a comma could be added between “Rhododendron” and “with” to make the sentence structure better balanced. Or please consider breaking it into two sentences for better readability. 

3.      “Azalea Lace Bug” should be lowercase, and “the” should be removed from the first sentence.

Introduction section

Major comments:

1.     The introduction starts by stating the significance of Rhododendron as an ornamental plant and introduces the azalea lace bug as a major pest. While this sets a general context, there could be a smoother transition between discussing Rhododendron’s importance and introducing the pest. 

2.     The identification of the gap in understanding the metabolic defense against pest stress in Rhododendron and the objective to address this gap using metabolomics is well-founded. 

Minor comments:

1.     The hybrid sentence “The Stephanitis pyriodes Scott, belongs to the family of Tingidae and is commonly known as the azalea lace bug (ALB), is one of the main pests…” is grammatically incorrect and should be revised for clarity.

2.     The term “realized to be inefficient” could be replaced with “found to be inefficient” for more academic tone. 

3.     “…diversed strategies” in the first sentence of the second paragraph should be “diverse strategies”.

Results section:

1.     The use of ten azalea lace bugs in the feeding preference test may be insufficient to yield robust results. The statement “6.1 out of total ten bugs” is unclear, particularly regarding how to interpret and apply an average of “6.1 bugs” in practical industry settings. To enhance the reliability of the findings, it is recommended to increase sample size to at least 50 bugs per replicate, which would provide more solid and interpretable results.

2.     In Figure 1, the term “parasitized on leaf” is not typically accurate or common when referring to the feeding behavior of the azalea lace bug, or most other insects that feed on plant tissues, especially considering that this was a 24-hour experiment whereas females deposit eggs over a 2-to-3-month period.

3.     Did the authors confirm the assumptions before using ANOVA?

Materials and Methods section

The logical sequence of experimental methods, starting from plant sourcing to complex bio-analytical techniques, is commendable. The inclusion of sophisticated analytical methods like HS-SPME and GC-MS, as well as HPLC-MS/MS, along with their parameters and conditions, provides a robust framework for chemical analysis and adds depth to the study’s methodological rigor. Here’s only one question the reviewer would like to ask: Given that the compounds released from leaf damage caused by piercing-sucking insects may differ from those released by mechanically cutting leaf tissues, how could the authors ensure that the volatile and non-volatile compounds emitted from cut leaf tissues of the two azalea cultivars and mixed in the same Petri dish do not interfere with the azalea lace bug’s behavior in the feeding preference assay described in sub-section 4.2.2? 

Reference section

The reference format in this manuscript deviated from the Plants guidelines. Please refer to the guidelines attached below and recent papers to polish the reference format.

The reference list should include the full title, as recommended by the ACS style guide. Style files for Endnote and Zotero are available.

References should be described as follows, depending on the type of work:

  • Journal Articles:
    1. Author 1, A.B.; Author 2, C.D. Title of the article. Abbreviated Journal Name YearVolume, page range.
  • Books and Book Chapters:
    2. Author 1, A.; Author 2, B. Book Title, 3rd ed.; Publisher: Publisher Location, Country, Year; pp. 154–196.
    3. Author 1, A.; Author 2, B. Title of the chapter. In Book Title, 2nd ed.; Editor 1, A., Editor 2, B., Eds.; Publisher: Publisher Location, Country, Year; Volume 3, pp. 154–196.
  • Unpublished materials intended for publication:
    4. Author 1, A.B.; Author 2, C. Title of Unpublished Work (optional). Correspondence Affiliation, City, State, Country. year, status (manuscript in preparationto be submitted).
    5. Author 1, A.B.; Author 2, C. Title of Unpublished Work. Abbreviated Journal Name year, phrase indicating stage of publication (submittedacceptedin press).
  • Unpublished materials not intended for publication:
    6. Author 1, A.B. (Affiliation, City, State, Country); Author 2, C. (Affiliation, City, State, Country). Phase describing the material, year. (phase: Personal communication; Private communication; Unpublished work; etc.)
  • Conference Proceedings:
    7. Author 1, A.B.; Author 2, C.D.; Author 3, E.F. Title of Presentation. In Title of the Collected Work (if available), Proceedings of the Name of the Conference, Location of Conference, Country, Date of Conference; Editor 1, Editor 2, Eds. (if available); Publisher: City, Country, Year (if available); Abstract Number (optional), Pagination (optional).
  • Thesis:
    8. Author 1, A.B. Title of Thesis. Level of Thesis, Degree-Granting University, Location of University, Date of Completion.
  • Websites:
    9. Title of Site. Available online: URL (accessed on Day Month Year).
    Unlike published works, websites may change over time or disappear, so we encourage you create an archive of the cited website using a service such as WebCite. Archived websites should be cited using the link provided as follows:
    10. Title of Site. URL (archived on Day Month Year).

See the Reference List and Citations Guide for more detailed information.

Comments on the Quality of English Language

Minor editing might be needed.

Author Response

(The authors gave the same response as above.)

Round 2

Reviewer 2 Report

Comments and Suggestions for Authors

The authors substantially improved the text for readability, clarity, and coherence. They addressed all criticisms and enhanced the methods, results, discussion, and conclusion.

Author Response

Dear Reviewer,

Thank you very much for your constructive comments and for the time you have invested in reviewing our manuscript. We are grateful for the positive feedback and are pleased to know that the revisions have significantly enhanced the readability, clarity, and coherence of our text.

We believe that these revisions have addressed the concerns raised and have further strengthened the contribution of our work to the field. We have also made sure to clarify any ambiguities and to present our findings in a manner that is both accessible and engaging to our readers.

We appreciate your expertise and the guidance you have offered. We hope that you find the updated manuscript satisfactory and that it meets the high standards of the journal.

Thank you once again for your valuable input.

Sincerely,

Jing Mao

Wuhan Academy of Agricultural Science,

Wuhan, 430075,

P.R. China

Tel: +86 13545001060

Fax: +86 27 87518860

Reviewer 3 Report

Comments and Suggestions for Authors

The reviewer acknowledges the authors’ significant revision in this version of their manuscript, and they responded to most of the comments well. There is only one question regarding their response to the comment: Did the authors confirm the assumptions before using ANOVA?

The reviewer notices that the authors’ statement regarding the assumptions tested before performing the ANOVA is incomplete. The statement accurately mentions using K-S and Shapiro-Wilk tests for assessing normality of distribution, with p > 0.05 suggesting no significant deviation from normality. However, 1) it is not necessary to use both tests for normality testing; typically, using one well-chosen test is sufficient. Please clarify. 2) these tests alone don't’ conclusively prove normality; visual inspection methods (e.g., Q-Q plots, histograms) should complement statistical tests for a comprehensive assessment about the normality. Please clarify. 3) A critical omission is the lack of mention of the homogeneity of variances assumption. This assumption is equally crucial for ANOVA and should be tested such as Levene’s test or Bartlett’s test. Please clarify.

Author Response

Dear Reviewer:

Thank you for your insightful comments and for acknowledging the significant improvements we have made in our revised manuscript. We appreciate your thorough review and have carefully considered your questions regarding the assumptions tested before performing the ANOVA.

We have provided a detailed point-by-point response to the your comments. Please see the attachment.
